# Protocol for The Toxin Study: Understanding clinical and patient reported response of children and young people with cerebral palsy to intramuscular lower limb Botulinum neurotoxin-A injections, exploring all domains of the ICF. A pragmatic longitudinal observational study using a prospective one-group repeated measures design

Lesley R Katchburian [1,2] Kate Oulton,[3] Eleanor Main [2] Christopher Morris [4] Lucinda J Carr [1]

For numbered affiliations see end of article.

**Correspondence to**
Lesley R Katchburian;
lesley.katchburian@gosh.nhs.uk

## ABSTRACT

**Introduction** Botulinum neurotoxin-A (BoNT-A) is an accepted treatment modality for the management of hypertonia in children and young people with cerebral palsy (CYPwCP). Nevertheless, there are concerns about the long-term effects of BoNT-A, with a lack of consensus regarding the most meaningful outcome measures to guide its use. Most evidence to date is based on short-term outcomes, related to changes at impairment level (restrictions of body functions and structures), rather than changes in adaptive skills (enabling both activity and participation). The proposed study aims to evaluate clinical and patient reported outcomes in ambulant CYPwCP receiving lower limb BoNT-A injections over a 12-month period within all domains of the WHO's International Classification of Functioning, Disability and Health and health-related quality of life (HRQoL).

**Methods and analysis** This pragmatic prospective longitudinal observational study will use a one-group repeated measures design. Sixty CYPwCP, classified as Gross Motor Function Classification System (GMFCS) levels I–III, aged between 4 and 18 years, will be recruited from an established movement disorder service in London, UK. Standardised clinical and patient reported outcome measures within all ICF domains; body structures and function, activity (including quality of movement), goal attainment, participation and HRQoL, will be collected preinjection and at 6 weeks, 6 months and up to 12 months postinjection. A representative subgroup of children and carers will participate in a qualitative component of the study, exploring how their experience of BoNT-A treatment relates to clinical outcome measures.

## Strengths and limitations of this study

► To our knowledge, this is the first published use of the 'Quality Function Measure' to objectively evaluate quality of movement (QoM) following Botulinum neurotoxin-A (BoNT-A) treatment and explore whether any change in QoM translates into changes in outcome within activities, participation and health-related quality of life.

► This pragmatic mixed methods observational study will evaluate response to lower limb BoNT-A injections in ambulant children and young people with cerebral palsy using standardised outcome measures within all domains of the International Classification of Functioning, Disability and Health (ICF) over a 12-month period.

► This study is designed to optimise targeting of BoNT-A treatment by identifying patterns of response to BoNT-A treatment across all ICF domains in order to assist clinicians and families in making informed decisions about future treatment.

► This is a pragmatic one-group study design, conducted from a single tertiary (specialist children's) centre. Participants will continue to receive their routine rehabilitation therapy and orthotic provision at local community level. Individualised intervention parameters may make study replication difficult in other settings.

**Ethics and dissemination** Central London Research Ethics Committee has granted ethics approval (#IRAS 211617 #REC 17/LO/0579). Findings will be disseminated in peer-reviewed publications, conferences and via networks to participants and relevant stakeholders using a variety of accessible formats including social media.

## INTRODUCTION

Cerebral palsy (CP) is the most common cause of physical disability in childhood.[1] Although the initial brain insult is described as static, the effects of the neurological involvement are dynamic and change with time and growth of the child.[2] Increased tone (hypertonia) is considered one of the primary motor impairments in children and young people with cerebral palsy (CYPwCP)[3] and a significant contributor to secondary musculoskeletal impairments impacting on activity and participation.[4]

Since the 1990s, intramuscular Botulinum neurotoxin-A (BoNT-A) has become an internationally accepted treatment modality for the management of hypertonia in overactive muscle groups.[5] BoNT-A once injected into the hypertonic muscle produces a 'reversible' temporary localised muscle weakness by blocking acetylcholine release at the neuromuscular junction. The pharmacological effect is said to last for 12–16 weeks, and the ability of BoNT-A to reduce focal hypertonia in ambulant CYPwCP has been well documented.[6–8] While the effects can be observed within 24–72 hours following injection, the period of clinically useful relaxation is reported to last between 3 and 6 months.[9 10]

The progression of dynamic contracture to fixed contracture is a fundamental issue in the care of the child with CP. The period of decreased hypertonia following BoNT-A provides a 'window of opportunity' for therapy to address specific predetermined goals of rehabilitation, such as stretching and strengthening of muscles, increased range of motion of joints, improved postural management and pain relief. Despite the temporary effect of injections, gains in motor function have been reported to last as long as 12 months.[11]

Several studies have demonstrated the benefits of BoNT-A injections for ambulant CYPwCP in Gross Motor Function Classification System (GMFCS) levels I–III,[12] particularly at single level use (injection at one level; eg, the calf complex to treat equinus foot posture).[3 13–17] A meta-analysis of double-blind, randomised controlled trials (RCTs) confirmed superiority of BoNT-A over placebo injections into the calf complex on improvement of gait in patients with spastic equinus.[18] BoNT-A treatment, in combination with other rehabilitation treatments, has resulted in a significant improvement in functional goal attainment over and above those in a non-BoNT-A treatment group,[19] with improvements in gait parameters, pain reduction and splint tolerance.[20–23]

Nevertheless, despite positive outcomes for single level use, RCTs of the effectiveness of multilevel use of BoNT-A injections report mixed results.[24 25] Although a recent systematic review of interventions for CYPwCP, identified BoNT-A treatment as one of few interventions

with a sound evidence base[26], a recent Cochrane Collaboration report by Blumetti *et al*[27] showed less favourable results. The report reviewed 31 studies, assessing 1508 participants and concluded that there is limited evidence to show that BoNT-A, when compared to placebo or usual care, improves walking, joint motion, satisfaction with outcome of treatment and muscle spasticity in CYPwCP. Sample sizes in BoNT-A studies are often small and predominately based on short-term outcome (3–6 months) with few assessing outcomes beyond 6 months.

Although it is widely acknowledged that BoNT-A treatment is not a 'stand-alone' treatment, detailed information regarding the adjunctive measures used in conjunction with BoNT-A is often lacking, making evaluation of its efficacy difficult.[28–30] Some authors have highlighted the difficulty in relating changes in impairment measures following BoNT-A treatment to functional improvements in CYPwCP, with little reference made to minimal clinically important differences (MCIDs).[24] This raises concerns that current standardised outcome measures, focusing predominantly on impairment measures (without relating this to other domains of the ICF), may lack the sensitivity to pick up meaningful changes following injections, or indeed overestimate treatment effects if these do not relate to MCIDs, all highly pertinent when planning repeat treatment.[24 27 31]

Recent studies investigating pathophysiological changes within hypertonic muscle have highlighted potential histological changes following both single and repeated BoNT-A treatment.[32] A number of authors have suggested potential harm following repeated BoNT-A use.[33–36] However, both positive and negative effects have been reported, and a variability in measurement techniques and muscles assessed makes comparison between studies challenging.[37–42] Although BoNT-A has been described as a 'reversible treatment',[43] some authors suggest that BoNT-A exposure in CYPwCP may be associated with impaired muscle growth in the short-term[44–47] and potential long-term atrophy.[41 45 48]

The WHO's International Classification of Functioning, Disability and Health (ICF) encourages evaluation of adaptive skills (enabling activities and participation) and health-related quality of life (HRQoL) in order to target interventions that are meaningful to CYPwCP and their families (figure 1). However, evidence for BoNT-A treatment remains mostly related to measures of body functions and structures, and less evidence pertains to activity and participation.[22 25 49–51] Few trials have explored improvement in the activity and participation domains or HRQoL after BoNT-A injections,[24 52] and qualitative data relating to CYP and caregivers experience are rarely incorporated.[53–55]

Longitudinal changes are not well characterised, and evidence of impact on CYPwCP is elusive. The uncertainty that exists around BoNT-A treatment has resulted in a call to extend the period that CYPwCP are followed up after BoNT-A treatment beyond the short-term, 12-week period,[44] with an imperative to evaluate interventions

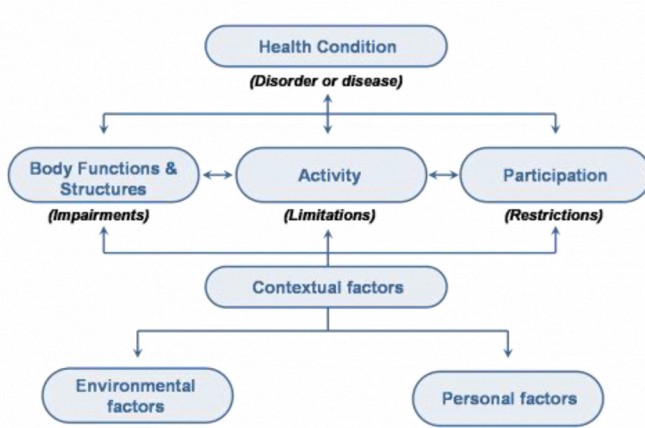

**Figure 1** WHO's International Classification of Functioning, Disability and Health schematic representation of living with a health condition.

with BoNT-A using more sensitive outcome measures that evaluate meaningful aspects of health and quality of life for CYPwCP and families.[35 44 56–58]

Concern has been raised that BoNT-A treatment may be overprescribed for CYPwCP[35 36 38] if there is little guidance about which patients will benefit, risking potential harm if outcomes are unclear.[38 59–61] While clinical evidence suggests that BoNT-A remains a valuable treatment option,[15 58 62 63] there is a need to optimise its use by developing clear guidelines and robust treatment algorithms in order to predict which children and young people will benefit from the addition of BoNT-A intervention to their overall management programme and when it is preferable to consider other treatment options.

### The Toxin Study
This paper describes the protocol for a pragmatic prospective longitudinal observational study in an established paediatric movement disorder service in London, UK. As BoNT-A treatment is considered best practice care for focal hypertonia management in CP,[19 21 48] there are practical and ethical concerns regarding the inclusion of a 'no treatment' control group and so a comparator was deemed not ethically appropriate.

The aims of this study are to:
1. Investigate clinical and patient reported outcomes (of body structures and function, quality of movement, activities and participation and HRQoL) associated with lower limb BoNT-A injections in ambulatory CYPwCP over a 12-month period.
2. Determine any factors associated with a response to BoNT-A treatment.
3. Explore qualitatively how standardised clinical outcome measures relate to the experiences of CYPwCP following BoNT-A treatment.

### METHODS
### Study design
This is a mixed methods study comprising of two phases:

Phase 1: to meet objectives 1 and 2, we will use a prospective longitudinal study using a one-group repeated measures design with each child acting as their own control.

Phase 2: to meet objective 3, we will conduct interviews with a subgroup of CYPwCP and parent/carers from phase 1 to elicit their experiences and views of change following BoNT-A treatment.

Using a convergent mixed methods approach, quantitative data from phase 1 will be synthesised with qualitative data from phase 2 in order to gain understanding of the impacts of BoNT-A treatment.[55 64]

### Study sample and recruitment
Potential study participants will be identified from clinical lists of the Movement Disorder Service at Great Ormond Street Hospital for Children (GOSH). All eligible participants will be invited to take part in phase 1 and will be enrolled sequentially. A subgroup of CYPwCP and their parents/primary carers will then be invited to participate in phase 2 following review at 6 months postinjection. Purposive sampling will be used to ensure a representative sample of CYPwCP within each GMFCS level.

### Inclusion and exclusion criteria
Participants with a confirmed diagnosis of CP meeting the following criteria will be included: (1) ambulant, functioning at GMFCS levels I–III; (2) aged between 4 and 18 years; and (3) prescribed lower limb BoNT-A injections in their clinical management plan for dynamic hypertonia limiting functional goals or causing pain.

Children will be excluded if they have previously had: (1) orthopaedic surgery to the injected muscle; (2) neurosurgery for tone reduction (selective dorsal rhizotomy); (3) lower limb BoNT-A injections in the last 6 months, or currently have (4) unrelated musculoskeletal problems such as recent acute injury or congenital structural abnormality; (5) no access to a block of therapy (defined as a minimum of 6 weekly sessions) postinjections; and (6) an inability to complete baseline assessments due to capacity, ability or willingness.

Parents: every effort will be made to support the inclusion of all families invited to participate in the study (including those where English is not their first language). We will use translators when required to ensure that there is sufficient understanding to complete the measures.

### INTERVENTION
### Motor assessment
An experienced multidisciplinary team (consisting of a consultant paediatrician/neurologist and senior physiotherapist) will identify muscle groups to be injected. Muscle selection will vary between participants according to the presence of dynamic hypertonia (based on clinical assessment with reference to standard definitions[65]), related functional impairment and individual goals of the CYPwCP. All participants will receive a 6-week block

 

of therapy post injection from their local therapy team, delivered locally in the community as per usual care. Details of dosage including frequency, location and type of therapy (eg, goal directed therapy and strengthening) together with participation activities will be recorded including any additional intervention such as casting or change in orthotic provision. This will be required in order to describe the content and parameters of usual care in as much detail as possible.[66]

### Administration of BoNT-A

We will follow standard clinical practice prescription of BoNT-A at GOSH which involves administration of 500U Dysport (abobotulinumtoxinA, Dysport Ipsen Ltd) diluted in 1ml of normal saline, up to a maximum dose of 30 units/kg/body weight or a total dose up to a maximum of 1000 units per injection session. All CYPwCP will continue to receive BoNT-A injections under ultrasound guidance as a day case, either under sedation with local analgesia or under general anaesthetic. Adverse events will be recorded, and standard reporting of dose per muscle and follow-up will be as per current clinical policy.

### Training and fidelity

Clinical staff collecting study data are experienced members of the tertiary service with extensive experience of working with CYPwCP (mean 21 years, range 13–33 years). A standardised protocol for measurement and documentation is used in the clinic, and an additional study manual with instructions for clinicians will be used to ensure consistency during the study. Two half-day training sessions will be provided for clinicians collecting study data prior to the start of recruitment. Monthly meetings with clinicians by the research team will ensure consistency and adherence to study protocol.

Staggered patient recruitment and data collection will commence in September 2017, and each participant will be followed up for a period of 12 months (figure 2). Data collection and final analysis of participant data are expected to be completed by September 2021. All decisions regarding clinical care, assessment frequency and BoNT-A injections will continue as per usual clinical practice. Standardised clinical assessments and outcome measures will be performed at four time points $T_0$–$T_3$. The timings and rationale for these are summarised in table 1

All participants *who have not undergone a surgical procedure* will be assessed at $T_3$, independent of outcome at $T_2$. This will facilitate analysis of factors associated with changes in impairment, activity, participation and HRQoL following BoNT-A treatment and evaluate time to reinjection over 12 months. The need for reinjection will be determined by clinical examination (documentation of a technical response, eg, change in MTS), evaluation of goal (Canadian Occupational Performance Measure) scores and in consultation with families and local team as per usual clinical practice.

### Validated outcome measures for all ICF domains

The standardised outcome measures used in the study are summarised within the ICF domains in figure 3 with administration details of the measures summarised in table 2. Outcome measures follow GOSH standard clinical practice with primary outcome measures marked in italics.* Patient assessment takes between 60 and 90 min.

### Classification of the participants

#### Gross Motor Function Classification System – Expanded and Revised (GMFCS-E&R)

The GMFCS-E&R[12] is an internationally recognised five-level system to classify the motor abilities of CYPwCP aged 4–18 years, with level I CYPwCP being identified as the most physically able and level V as the least. It is valid and reliable and frequently used in both research and clinical practice. Only ambulant CYPwCP classified as GMFCS levels I–III will be included in this study.

#### Classification of CP

Participants are classified according to the distribution (unilateral or bilateral) and tone presentation (hypertonia will be identified as predominantly spastic, dystonic or mixed in type) as identified by the guidelines of the surveillance of cerebral palsy in Europe network (SCPE).[67]

### OUTCOME MEASURES

### Primary outcome measures

*Quality Function Measure (QFM)* is an observational criterion referenced measure designed to evaluate the quality of movement in standing and walking skills in CYPwCP.[68] It is used in conjunction with the Gross Motor Function Measure (GMFM) using dimensions D and E, which focus on 'standing' and 'walking, jumping and running skills', which are considered proxy clinical gait measures. The GMFM is the 'gold standard' tool for evaluating gross motor function in CYPwCP, evaluating 'how much' of a gross motor skill a child can perform.[69] However, there are concerns regarding GMFM's sensitivity in capturing subtle yet meaningful change postintervention, due to 'ceiling effects' of the measure when used with ambulant CYPwCP in GMFCS levels I–III.[22 68 70] The QFM scores movement quality and assesses 'how well' a child performs gross motor tasks.[68] It has shown excellent rater and test–retest reliability (Intraclass correlation (ICC) 0.89–0.97). Minimal detectable change estimates (6%–9%) suggest that the scale has potential as an evaluative measure (V Wright, personal communication, 8 May 2018). However, to date, there are no published studies evaluating the responsiveness of QFM following BoNT-A injections.

In order to minimise bias, the video analyst will be blinded to the stage of treatment and assessment time point (ie, pre $T_0$ or post $T_{1-3}$ injection). To conceal these time points, each video containing GMFM D and E (standing and walking) dimensions is anonymised and randomly allocated a letter by a coworker not involved in the service. To minimise recall bias, a time lag will be

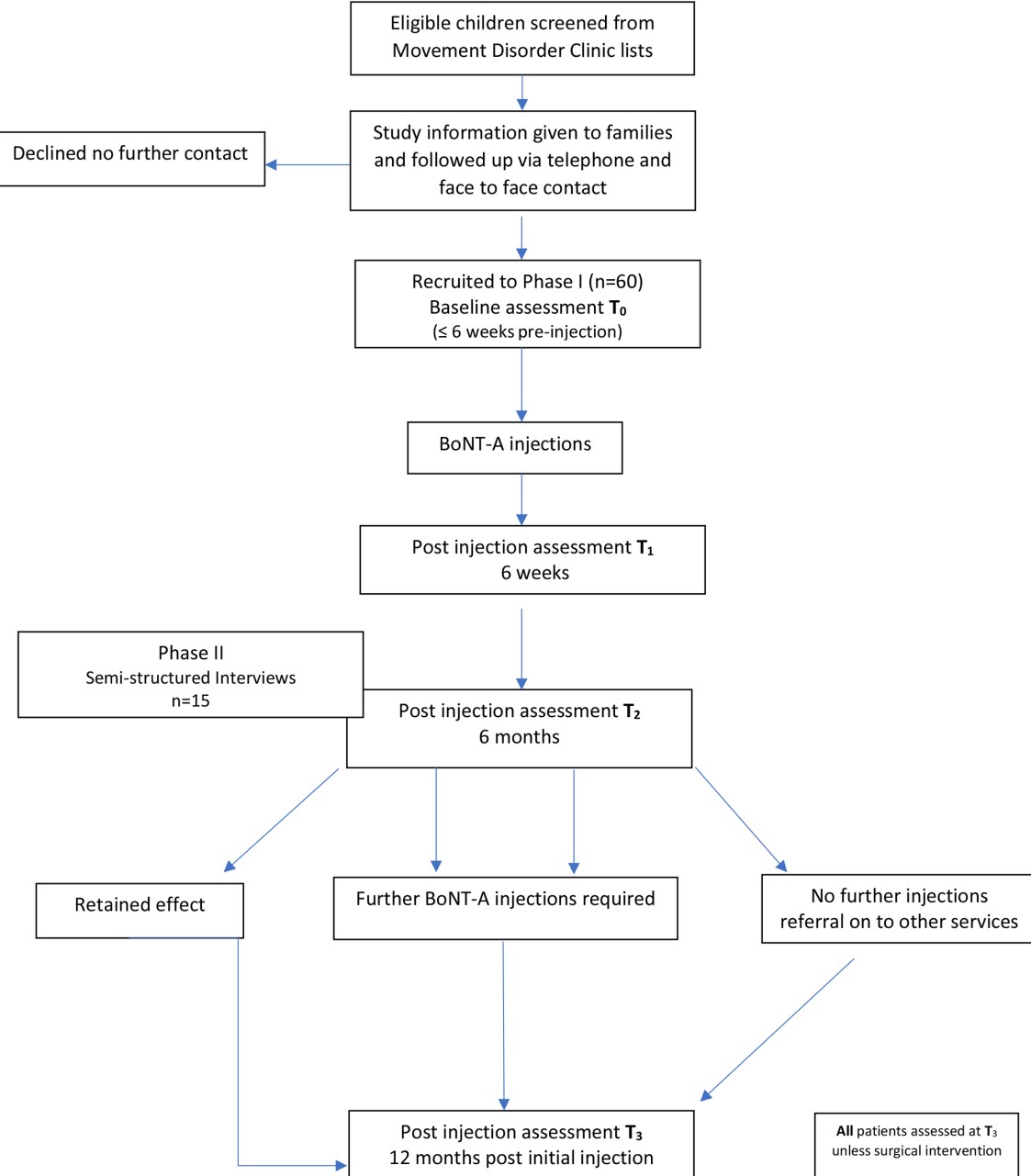

**Figure 2** Study design flow chart of patient recruitment and data collection during the study. BoNT-A, Botulinum neurotoxin-A.

built in of 2 weeks and videos of at least 10 other children will be scored before scoring the video of the same child at a different assessment time point. This is in keeping with previous QFM reliability studies that suggest a gap of 2 weeks before evaluating test–retest scores.[68 70] Data will be entered into a secure database without access to previous scores until scoring for all assessment time points is complete.

*COPM* is a goal attainment tool modified for use in the paediatric population and frequently used in neuro-disability research.[71–73] It identifies concerns regarding 'occupational performance issues', that is, the ability to carry out functional tasks, allowing the identification of goals and has been used to document change post BoNT-A rehabilitation.[74] In the paediatric population,

areas of concern in a child's self-care, activity and leisure are explored during the preassessment appointment with the clinical team. COPM has demonstrated high retest reliability (ICC 0.76–0.89), sensitivity to change and good content, construct and criterion validity for CYPwCP receiving BoNT-A.[75 76] Families are asked to identify up to three areas of concern that they and their child hope to improve following lower limb BoNT-A injections. Whenever possible, goals are set with the child's input.[77] The child and parents are asked to rate their perception of both current performance and their satisfaction with this performance on a 1–10 ordinal scale. A score change of two or more points is considered clinically significant.[72 78]

**Table 1** Timings of study assessments $T_0$–$T_3$

| | | |
|---|---|---|
| $T_0$ | Preinjection baseline measures | 1–6 weeks before injection |
| $T_1$ | 6 weeks postinjection | Estimated time to reach target threshold for BoNT-A. 'Evaluation of efficacy of injections'. |
| $T_2$ | 6 months following injection | Expected completion of pharmacological action. 'Evaluation of retention of effects post injection'. |
| | At $T_2$, as per usual clinical practice, there are three possible outcomes for participants:<br>▶ Favourable response to injections with retention of effects – no further injections indicated at this time.<br>▶ Favourable response to injections – listed for a second injection cycle.<br>▶ Non favourable response to injections – discharged to other services (eg, neurosurgery/orthopaedics). | |
| $T_3$ | 12 months following initial injection | End of study |

### Secondary outcome measures

*Modified Tardieu Scale (MTS)* is a recognised clinical measure to differentiate dynamic spasticity from fixed contracture in a muscle. It determines the passive range of movement at two different movement velocities fast (R1) measuring dynamic muscle length and slow (R2) measuring static muscle length, with the relative difference between the two (R2–R1) determining the dynamic tone component of the muscle contracture.[79 80] It is measured in degrees with a universal goniometer using standardised testing positions for each muscle. It has been postulated that the larger the dynamic tone component, the more amenable it is to treatment with BoNT-A.[78] It is more effective than the Modified Ashworth Scale in identifying the presence of spasticity (88.9%, kappa=0.73, p=0.000) and the presence of contracture (77.8%, kappa=0.503, p=008).

*Faces Pain Scale* is a self-report measure of pain intensity developed for children, which has been shown to have good psychometric properties for pain reporting (modified for use by carer when the child is unable to self-report). The scale shows a close linear relationship with visual analogue scales. It is measured by an ordinal scale from 0 to 10 with pictorial representation of faces. A change score of 2 or more is said to be clinically significant.[81]

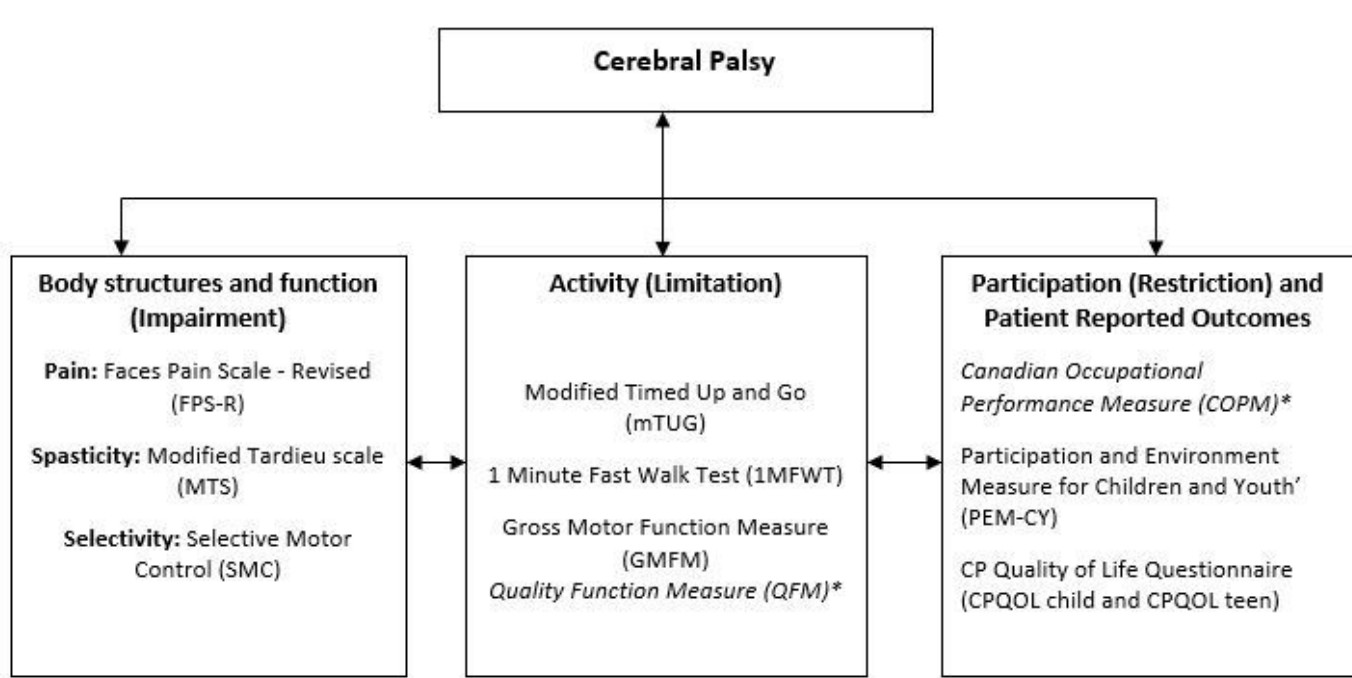

**Figure 3** Schematic representation of ICF domains including standardised outcome measures used in the study. ICF, International Classification of Functioning, Disability and Health.

**Table 2** Summary of outcome measures used in the study

| ICF domains | Assessment | Outcome measures | Method of administration | Units | Description |
|---|---|---|---|---|---|
| Body functions and structures (impairment) | Hypertonia and dynamic range of movement | Modified Tardieu Scale (MTS) | Standardised goniometry placement | Degrees | MTS measured at injected muscles. The difference between the slow stretch R2 and a fast stretch R1 is reported as the 'dynamic range' and when a difference is present, this 'dynamic range' is reported to be amenable to treatment with BoNT-A. |
| | Pain | Faces Pain Scale (FPS-R) | Score assigned by CYPwCP | Score 0–10 | FPS-R has been shown to have good psychometric properties for pain reporting (modified for use by carer when CYPwCP unable to self-report) |
| | Selective motor control | Selective Motor Control scale (SMC) | Score assigned by clinician | Score 0–4 | SMC of the ankle is assessed using a standardised test procedure. |
| Activity (functional limitation) | Gross motor function | Gross Motor Function Measure (GMFM) (D&E walk/run/jump dimensions) | Video recorded Standardised assessment form Scored by clinician | % score | The GMFM is designed to evaluate change in gross motor function over time in CYPwCP. GMFM is considered the standard outcome assessment tool for clinical intervention in CP, dimensions D&E will be used and are considered a Proxy Functional Gait Measure. *Families are consented for video storage in accordance with the Great Ormond Street Hospital policy.* |
| | Quality of Gross Motor Function | *Quality Function Measure (QFM)* | video scored later by Prinicpal Investigator (LRK) blinded to treatment stage | % score for five quality attributes | The *QFM* is an observational validated measure that captures the quality of movement of the items in the GMFM (D&E dimensions) This is scored from GMFM video involving no extra assessment time for CYPwCP. |
| | Balance/functional mobility | Modified Timed up and Go Test(mTUG) | Timed standardised test (from sitting CYP stands and walks distance 3 m touches star returns to seat) | Seconds | mTUG integrates transitions and walking skills and provides a meaningful measure of capability. It has been shown to be a reliable outcome measure for assessing functional mobility in CYPwCP. Proxy Functional Gait Measure. |
| | Walking ability(Efficiency) | 1 min fast walk test (1MFWT) | Distance recorded (5 min rest followed by walking for 1 min in a 9 m corridor at maximum walking speed without running. CYPwCP permitted to use normal walking aids and orthoses). | Metres | 1MFWT is a good discriminator of functional ability for dynamic balance, muscle performance and endurance. Proxy Functional Gait Measure |

Continued

**Table 2** Continued

| ICF domains | Assessment | Outcome measures | Method of administration | Units | Description |
|---|---|---|---|---|---|
| Participation (restriction) | Involvement in daily activities | Participation and Environment Measure for Children and Youth' (PEM-CY) | **Parent-reported questionnaire** (25 item ~25 min to complete) *Answered at home or in clinic* | Summary score | The PEM-CY assesses participation frequency and involvement in home school and community, along with environmental factors within these settings. Can be completed online at home or in clinic on a handheld device/paper format while waiting for the child's assessment to be completed. |
| Health-related quality of life | Quality of life across seven domains | Cerebral Palsy Quality of life measure (CPQOL) | CYP (or proxy) reported Questionnaire CPQOL-Child/CPQOL Teen *Answered at home or in clinic* | Mean Domain Score | The CPQOL-Child is designed to assess condition-specific quality of life of children across seven domains for children aged 4–12 years – parent report for children aged 4–12 years and a self-report for children aged 9–12 years (52–66 items). CPQOL-Teen 13–18 years. Adolescent self-report version and a parent version (72–89 items). |
| Goal setting (ICF domains) | Selection of goals CYP hope to improve following | *Modified Canadian Occupational Performance Measure (mCOPM)* | In clinic three goals set by CYP and family preinjection Scores assigned postinjection | Score (1–10) | *mCOPM* rates perception and satisfaction of CYP's performance on a 1–10 ordinal scale. A score change of >2 points is considered clinically significant. |

BoNT-A, botulinum toxin-A; CYPwCP, children and young people with cerebral palsy; ICF, International Classification of Functioning, Disability and Health.

*Selective Motor Control* is a good discriminator of selective motor control and assesses a child's ability to voluntarily and selectively control the dorsiflexors of the ankle.[77] It is measured by an ordinal scale between 0 and 4. Higher score indicates better selectivity, inter-rater agreement has been shown to be moderate to good (kappa=0.58–0.77) with strong test–retest reliability (kappa=0.88–1).[79 82]

*One Minute Fast Walk Test* has been chosen as a proxy gait measure and is a good discriminator of functional ability for dynamic balance, muscle performance and endurance.[83] Participants walk for 1 min along a 9 m corridor at maximum speed without running. They are able to use their usual walking aids and orthoses. Distance is calculated to the nearest 10 cm. It shows concurrent validity with the GMFM, with significant correlation between GMFM score and distance walked (r=0.92) and good reliability (ICC: 0.97).[84]

*Modified Timed Up and Go* is a good discriminator of balance, anticipatory postural control and functional mobility in CYPwCP. Participants are timed rising from a chair, walking 3 m to touch a star on the wall and returning to sit down as quickly as they can without running. Participants are able to use their usual walking aids and orthoses. The measure differentiates performance between CYPwCP at GMFCS levels I–III. It has shown excellent inter-rater reliability (ICC 0.83–0.99)[85]

*Participation Environment Measure – Child and Youth (PEM-CY)* is an innovative parent-reported participation measure for use with CYP between 4 and 18 years. It measures CYPwCP's participation at home (10 items), school (5 items) and community (10 items), taking into account the environmental challenges within each setting. Summary scores for participation frequency, child's involvement and parental desire for change are calculated for all domains. PEM-CY has demonstrated good internal consistency (ICC 0.72–0.83) and test–retest reliability (ICC 0.76–0.89).[86]

*Cerebral Palsy Quality of Life Measure (CPQOL)* is an HRQoL assessment administered by questionnaire, specifically designed for CYPwCP. It quantifies 'well-being' across seven key HRQoL domains. Items are scored on a nine-point rating scale, then summed and averaged to generate seven domain scores. There are two child-reported (CPQOL-Child: 9–12 years and CPQOL-Teen: 13–18 years) and two proxy-reported versions dependent on child's age and cognitive ability (CPQOL-Child Primary caregiver: 4–12 years and CPQOL-Teen Primary care giver: 13–18 years). The questionnaire designers have stipulated a minimum age of 9 years for child self-reporting, however significant disagreement has been reported between child and parent proxy reports in many HRQoL instruments, and the child's perspective will be sought whenever possible in this study.[87] The CPQOL has demonstrated good internal consistency (child ICC 0.74–0.92; teen 0.81–0.96) and adequate test–retest reliability (child ICC 0.76–0.89; teen 0.59–0.83).[88 89]

## Semistructured interviews

Interviews will be used to elicit CYPwCP and parent/carers views of change following BoNT-A treatment. These are expected to last 60 min for parents and up to 30 min for children and will take place at home or in the hospital (depending on family preference). Interviews will be guided by a predetermined interview schedule, which will explore perceived change across all ICF domains to facilitate comparison with quantitative outcome measures, as well as investigate the acceptability of the standardised outcome measures used inphase 1 of the study. Parents will be interviewed separately from CYPwCP, although individual preference for parents to be present or absent during interviews with their child will be respected. Interviews with CYPwCP will be tailored to their age, cognitive and communication ability using a variety of different techniques, including art-based activities. Offering a toolkit of different creative techniques will ensure that the activities will be accessible to all CYPwCP involved in the interviews, acknowledging their different skills and personalities and their cognitive and physical abilities that is particularly important in this population of CYP.[90] To ensure a representative sample of participants for phase 2, CYPwCP with a good, moderate and poor response to toxin (determined by the clinical team at the 6-month assessment) within each GMFCS level (I–III) will be invited to take part in phase 2. All interviews will be audio recorded with permission from participants.

## Sample size

Phase 1: this study has been powered to detect a difference on one of the two primary outcome measures, the COPM. The sample size power calculation is based on anticipated change in the COPM goal performance at the primary end point T1 (6 weeks postintervention) after BoNT-A treatment. A change in score of 2 or more points on the performance scale of the COPM would be considered clinically meaningful.[72 78] An earlier study of lower limb BoNT-A intramuscular injections yielded SD between 1.4 and 1.7 for COPM performance.[77] Based on a conservative estimate using a mean change of two points on the COPM performance scale (power 0.8, two tailed, $p<0.05$), 36 participants (12 in each GMFCS level) are required. Allowing for attrition and missing data over a 12-month period, a total of 60 participants (20 in each GMFCS level) will be recruited for phase 1 of the study.

As there are no studies reporting Quality Function Measure (QFM)[68] as a primary outcome measure following BoNT-A injections, no data exist to inform a power calculation. It is anticipated that the results from this study will provide power calculation data for use in future multicentre trials using QFM.

Phase 2: a sample size of approximately 15 CYPwCP (five from each of the GMFCS levels I–III) and their parent/carer is anticipated to be sufficient to reach thematic saturation for the qualitative element of the study.[91]

## Data analysis

Descriptive analysis of the prospective study will provide information on baseline characteristics of impairment, activity, participation and HRQoL. Means and SD will be used for normally distributed data and medians and IQRs for skewed data. Other demographic data will be described in a similar way with frequencies and proportions used for categorical information. The primary outcomes of the intervention will be assessed using generalised estimating equations for longitudinal analysis to evaluate differences in continuous data at postintervention time points. Multilevel regression models will be used to investigate the importance of the various potential predictors over a 12-month period on the four main outcomes: impairment, activity, participation and HRQoL. First, univariate relationships will be explored. and then if appropriate, we will fit a multivariable model for each outcome.

All interviews will be transcribed verbatim, and transcripts will be checked by the interviewer. Qualitative data will be analysed using Braun and Clarke's six phase framework for thematic analysis figure 4.[92] Two researchers (LRK and KO) will undertake analysis of the transcripts to determine consistent themes and will then map these onto results from standardised outcomes. This will allow themes to emerge that can then be used to evaluate how closely standardised outcome measures relate to family experience.

## Participant and data management

Electronic data will be managed through a secure database held in GOSH. Participants will be allocated an identification code that will be used to deidentify their files and forms. Pseudonyms will be used in all reports and publications, and direct quotations will be anonymised. Paper documents and other manual files will be appropriately filed and stored securely in a locked filing cabinet at GOSH. Demographic information and consent forms will be stored separately to research data. Classification measures, child demographics and related information will be taken prior to baseline for the purposes of stratification and description of the sample. Audio and visual recordings will be uploaded onto secure password-protected encrypted National Health Service computers and deleted from the recording device immediately after uploading. This protocol has been reviewed and approved by the Research and Development team and Caldecott Guardian at GOSH.

## Patient and public involvement

Children and young people with CP and their parents have been involved in the development of this project, exploring the importance of the research, the appropriateness of the research questions, the acceptability of the research methods and best methods for disseminating findings. Fifteen ambulant CYPwCP receiving BoNT-A treatment at GOSH and their parents were consulted individually. A wider population of CYPwCP and their parents were also consulted via the SCOPE website and

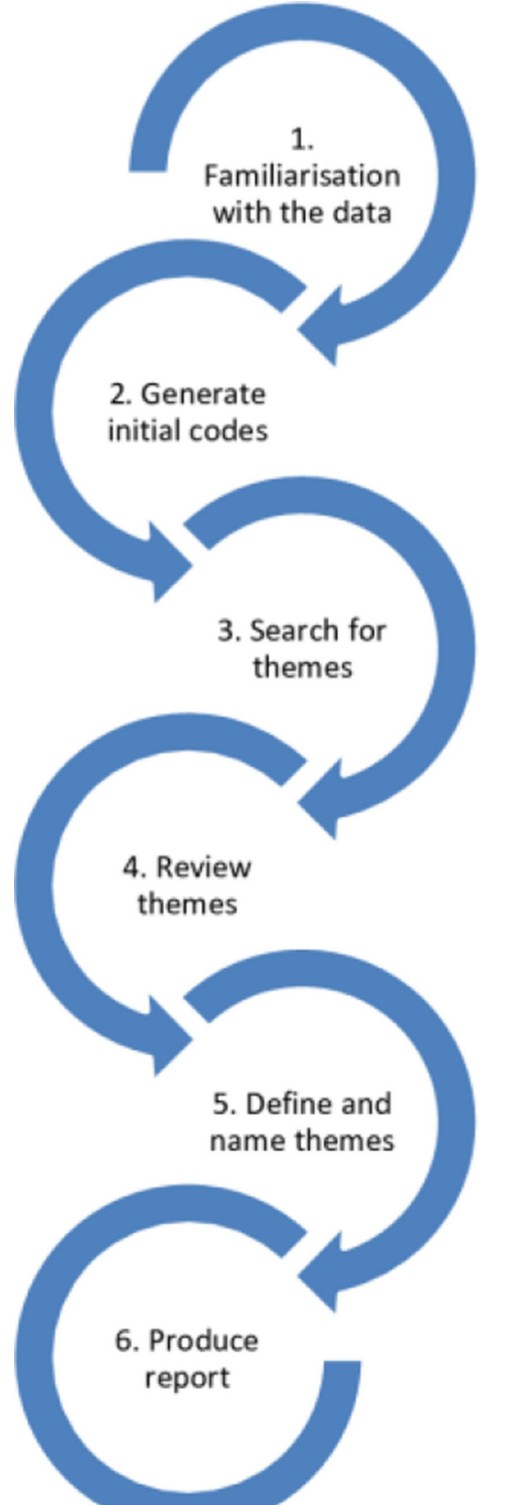

**Figure 4** Braun and Clarke's six-step approach to thematic analysis.

the advisory groups for CYPwCP at Brunel University and Young People's Advisory Group at GOSH.

### Reference group/steering group

Three parents (all mothers) and three CYPwCP (two boys and one girl (not participants)) continue advising on the study through membership of the study steering group (parents) or reference group (CYP). Three professionals (two physiotherapists and one doctor) are also part of the steering group. By including CYPwCP and families perspectives together with regular contributions from practising clinicians within the service, we hope to ensure the inclusion of important values and preferences from families and clinicians alike, ensuring that the findings of the study remain relevant and applicable to the management of ambulant CYPwCP.[24 93 94]

### DISCUSSION

This paper presents the protocol for a novel pragmatic prospective longitudinal observational study of BoNT-A use in ambulant CYPwCP at an established tertiary children's centre in the UK. It will measure outcomes across all domains of the ICF and HRQoL over a 12-month period to observe change and examine factors associated with positive or negative response to lower limb BoNT-A injections. The introduction of a standardised outcome measure, QFM, to evaluate any change in movement quality following BoNT-A treatment will further inform exploration of the relationship between tone reduction and changes in activity, as well as any influence on participation and HRQoL for our cohort of CYPwCP. Mixed methods research designs are considered by many to be essential in the study of therapeutic interventions in CP.[55] Including qualitative study data will ensure that the experiences of CYPwCP following BoNT-A injections are elucidated, considered and further understood.

This study, by identifying patterns of response to BoNT-A injections in key aspects of health across all the ICF domains, will provide clinicians and families with meaningful information to inform future treatment planning and optimise the use of BoNT-A in CYPwCP. Evidence will be generated about the appropriateness of the outcome measures used in detecting meaningful change after BoNT-A injections as well as their acceptability to CYP. In addition, the relationship between the standardised outcome measures used to capture treatment effect and the perception of outcome by children and their families will be investigated.

The results of this study will assist in the development of a pragmatic set of standardised clinical outcome measures (recognising minimum clinically important differences) in order to evaluate the effects of BoNT-A treatment in this cohort of patients. We hope this work will inform future evaluative research; working towards closer consensus on patient selection, injection frequency and consideration of how long treatment should continue, in order to assess the value of long-term use of BoNT-A treatment and its role in the management of CYPwCP.

A dissemination strategy has been devised to ensure the findings of this research are made widely accessible to families and professionals in order to maximise impact on the care of CYPwCP. Working in partnership with parent groups will strengthen our dissemination strategy, ensuring findings are shared in a variety of accessible formats reaching

a wide range of families, professionals, as well as academics and policy makers helping ensure the findings are translated into practice. We will disseminate the results of the study through international peer-reviewed journals and at national and international conferences. A social media strategy will also be developed to ensure dissemination of a plain language summary of findings.

**Author affiliations**
[1]Neuroscience Unit,The Wolfson Neurodisability Movement Disorder Service, Great Ormond Street Hospital For Children, London, UK
[2]Physiotherapy, UCL Great Ormond Street Institute of Child Health, London, UK
[3]Centre for Outcomes and Experience Research in Children's Health, Illness and Disability (ORCHID), Great Ormond Street Hospital for Children, London, UK
[4]Peninsula Childhood Disability Research Unit (PenCRU), University of Exeter Medical School, University of Exeter, Exeter, UK

**Acknowledgements** We would like to thank the clinical team at The Wolfson Neurodisability Movement Disorder Service at Great Ormond Street Hospital for Children (GOSH), especially, Dr Belinda Crowe, Xanthe Hodgson, Josephine Scerri, Emilie Hupin, Katharine Phillipps, Tessa Burnett, Sarah Foster, Agnieszka Rozborska and Stephanie Cawker. We would also like to thank Dr Jo Wray from the Centre for Outcomes and Experience Research in Children's Health, Illness and Disability (ORCHID) at GOSH for support and advice on the planned data analysis. Many thanks to Professor Roslyn Boyd, Scientific Director of the Queensland Cerebral Palsy and Rehabilitation Research Centre for her valuable suggestions regarding methodology and selection of outcome measures and Dr Margaret Mayston (UCL) for her early contribution to the preliminary study design. Most importantly, huge thanks to all the children and their families who have been involved in the development of the research idea, provided advice and input to the study protocol and those who have agreed to participate in the study.

**Collaborators** Belinda Crowe; Xanthe Hodgson.

**Contributors** All authors conceived and designed the study. LRK developed the protocol with input from all authors. LRK and LC are responsible for recruiting and supporting all the participants. LRK, EM and KO will conduct the data analyses. LRK and LC wrote the first draft of the manuscript; all authors reviewed this and have approved the final manuscript.

**Funding** Lesley Katchburian is funded by a National Institute for Health Research (NIHR) Clinical Doctoral Research Fellowship Award (ICA-CDRF-2015-01-037). The study is also supported by the NIHR GOSH BRC at UCL GOSH Institute of Child Health.Kate Oulton is an NIHR Senior Nurse and Midwife Research Leader.

**Disclaimer** The views expressed are those of the authors and not necessarily those of the NIHR or the Department of Health and Social Care.

**Competing interests** None declared.

**Patient and public involvement** Patients and/or the public were involved in the design, or conduct, or reporting, or dissemination plans of this research. Refer to the Methods section for further details.

**Patient consent for publication** Not required.

**Provenance and peer review** Not commissioned; externally peer reviewed.

**ORCID iDs**
Lesley R Katchburian http://orcid.org/0000-0002-7523-7083
Eleanor Main http://orcid.org/0000-0002-9739-3167
Christopher Morris http://orcid.org/0000-0002-9916-507X
Lucinda J Carr http://orcid.org/0000-0001-9238-108X

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
