## [Reviewer comments · BMJ Open]

ARTICLE DETAILS

TITLE (PROVISIONAL)	Protocol for The Toxin Study: Understanding clinical and patient reported response of children and young people with cerebral palsy to intramuscular lower limb Botulinum Toxin-A injections, exploring all domains of the ICF: A pragmatic longitudinal observational study using a prospective one group repeated measures design
AUTHORS	Katchburian, Lesley; Oulton, Kate; Main, Eleanor; Morris, Christopher; Carr, Lucinda

VERSION 1 – REVIEW

REVIEWER	Kerr Graham Murdoch Children's Research Institute (MCRI) Royal Children's Hospital, Melbourne, Australia. Previous recipient of research support from Allergan. Currently, Surgeon's Advisory Board, OrthoPediatrics
REVIEW RETURNED	03-Sep-2020

GENERAL COMMENTS	Review Thank you for the opportunity of reviewing Protocol for the Toxins Study: Understanding clinical and patient reported response of children and young people with cerebral palsy to intramuscular, lower limb Botulinum Toxin – A injections exploring all domains of the ICF: A pragmatic longitudinal observational study using a prospective one group repeated measures design. Abstract: Page 3, Lines 34-44: The influence of the ICF and ICF-CY have dominated recent research and many publications in the use of Botulinum Toxin in children with cerebral palsy. This is not a novel approach, and its value remains uncertain. Advantages include a more holistic view of the patient. Disadvantages include downgrading objective measures of gait and function in favour of subjective measures. A balance between objective and subjective outcomes is important. Methods and Analysis: Page 3, Lines 45-55: This Reviewer considers that the Study Design (Pragmatic prospective longitudinal observational study) is a weak design and will be unable to provide scientific answers to the critical questions. The changes in Gross Motor Function and gait function, as a result of gains in GMFM, in younger children, are an order of magnitude greater than the effects of injection of BoNT-A. As such parents and clinicians, will inevitably attribute gains in Gross Motor Function to the program of injections, when these are the result of the natural history. Even in older children,
---

	when there are no gains in Gross Motor Function, the supportive environment of being in a clinic with supportive clinicians, with repeated assessments of the child, provides a powerful therapeutic environment, which is likely to yield positive responses in areas such as HRQoL. These effects are a response to the therapeutic environment plus the well-recognised placebo effect from injecting Botulinum Toxin. Without a control group, I think that it will not be possible to assess the magnitude of these confounding effects. Page 4, Lines 12-16: I do not believe that this is a novel study, there are numerous published studies which have employed the ICF Framework following Botulinum Toxin injection. In addition, some of these studies have a much stronger (RCT) design. Lines 19-24: To the best of my knowledge the Authors' assertion that this is the first study to use QFM is correct. Lines 27-32: A laudable, aspirational goal for the study but doubtful of this can be realised. Lines 37-44: This is a serious weakness of the study. The SPACE BOP Study which separated out the effect of Botulinum Toxin from the general rehabilitation milieu, in an RCT design, found no added benefit from injections. Schasfoort F, Dallmeijer A, Pangalila R, et al. Botulinum toxin has no added therapeutical value or cost-effectiveness for gross motor function, everyday physical activity or quality of life when combined with intensive functional physiotherapy. Dev Med Child Neurol, 2015;57(S5):8-9. This excellent study, has a stronger design than the proposed study in that it is a randomised controlled trial. It explores many of the outcome measures proposed in this study but in the RCT format. This study found no added benefit from injections of BoNT-A. However, the weaker study design proposed by these authors may well find a false positive result. This will confuse rather than clarify the evidence base. Introduction: Page 5, Lines: 4-8: The crucial assertion that: "increased tone (hypertonia) is considered to be one of the primary motor impairments ..." is a strong and important statement. Significantly, there are no associated references. Clinicians focus too much on spasticity, when weakness, loss of selective motor control and sensory disturbance, are of far greater importance to daily functioning, in children with cerebral palsy. than spasticity. Lines 15-16: The idea that injection of BoNT-A is "reversible" is not correct, it is clearly not completely reversible. There are no animal or clinical studies in children to date which have convincingly demonstrated that the effects of a single injection of BoNT-A, ever wear off completely or that the injected muscle ever completely recovers. Lines 33-48: Rather than referencing, articles of variable scientific merit, the recent Cochrane Collaboration Report by Blumetti et al, would have been an excellent summary of the limitations of injection of BoNT-A. This report emphasises the small, temporary and inconsistent therapeutic effect, with poor to very poor supporting evidence. Line 48: The "Traffic lights paper" has been criticised in the literature, and is not an adequate substitute for much more rigorous, unbiased and scientifically credible evidence summaries. Blumetti FC, Belloti JC, Tamaoki MJ, Pinto JA. Botulinum toxin type A in the treatment of lower limb spasticity in children with cerebral palsy. Cochrane Database Syst Rev. 2019;10:CD001408.
--	--

	Lines 50-60: The Authors' list the weakness in many studies of BoNT-A treatment including single cycle of injections and short-term follow-up. Page 6, Lines 3-8: The Authors' are correct to indicate there are problems with current standardised outcome measures, stating that injections of BoNT-A fail to produce a positive change, at the level of one MCID. It is common practice to blame the outcome measure, when it is the intervention that is the weak point. The Authors' are not correct in stating that negative studies "focus predominantly on impairment measures". There are many studies using gold standard functional outcome measures including GMFM and 3-dimensional gait analysis, in which the response to BoNT-A injection is minimal or absent and below the MCID. Lines 11-23: This is a very important summary of the harmful effects of BoNT-A injection which need to be translated and incorporated in the Authors' protocol. We repeat at this point that there is no study, in animal models, or children with cerebral palsy which show that skeletal muscle ever recovers fully after a single injection of BoNT-A. Lines 25-37: The shifting emphasis from gold standard measures of gait and function, to "soft" outcome measures in activity measures and participation domains of ICF is regrettable. It reflects the negative outcomes of studies using the gold standard measures of function as much as a response to the prevailing orthodoxy that activities and participation are more important. Lines 40-47: It is perfectly reasonable to suggest outcome measures in all domains of ICF but this should not be at the expense of ignoring the gold standard objective outcome measures. Lines 50-58: This reviewer has mounting evidence that BoNT-A is overused to the detriment of long-term function in children with cerebral palsy. A single cycle of injections and study duration of 12 months is unable to address these key questions. Page 7, Lines 7-24: As stated above, this study has a weak design and will not achieve its desired goals of providing robust scientific evidence for the benefits versus harms for injection of BoNT-A. Line 57: The age range for inclusion is 4-18 years. We have previously reported no response to injections of BoNT-A in ambulant children with cerebral palsy, in a hemiplegic pattern by the age of 6 years because of the presence of fixed contracture. Hastings-Ison T, Sangeux M, Thomason P, Rawicki B, Fahey M, Graham HK. Onabotulinum toxin-A (Botox) for spastic equinus in cerebral palsy: a prospective kinematic study. J Child Orthop. 2018;12(4):390-7. In that study, children with diplegia had an "improvement" in equinus at the expense of increased crouch gait. Older children and adolescents, with cerebral palsy GMFCS Levels I-III, have a complex mixture of weakness, spasticity, lower limb contractures and bony torsion which are not responsive in any meaningful way to injections of Botulinum Toxin A. Many of these children require orthopaedic surgery, yet the presence of contracture, torsion, or joint instability are not listed here as exclusions to being involved in the proposed study. The protocol requires exclusions based on physical examination measures indicating the presence of muscle tendon contracture, bony torsion, and joint instability using clinical, radiographic and motion analysis data. Lines 26-40: Much more detail regarding the injection protocol needs to be incorporated here, not just referral to reference 61.
--	--

	Lines 45-55: The injection protocol is poorly described precluding replication or even understanding the connection between the indication and injection procedure. Page 9, Line 17-27: I note that the data collection “will commence in September 2017...expected to be completed in May 2021” as such, has this study started and is it more than half way to completion? Lines 30 onwards: The Table describing the assessments and outcomes, has multiple subjective terms, precluding scientific analysis and replication. Page 10, Lines 10: For the first time, it is acknowledged that some children will develop a contracture requiring surgical intervention. The identification of these children and exclusion from this protocol needs to be clearly articulated. We now see a constant procession of children presenting to our Gait Analysis Laboratory, with decompensated musculoskeletal pathology, far too late for effective treatment, because of persisting with ineffective injections of BoNT-A over many cycles. Line 26: “Patient assessment takes between 60-90 minutes”. This is burdensome for families for an intervention which the majority of current studies indicate will be ineffective. Page 11: The ICF Table of Outcome Measures: This is a strange melange of outcome measures not all of which relate to real world concerns of the parents of ambulant children with cerebral palsy GMFCS Levels I-III. The majority of parents, whose children are ambulant, have concerns about their gait pattern and their Gross Motor Function. Preston N, Clarke M, Bhakta B. Development of a framework to define the functional goals and outcomes of botulinum toxin A spasticity treatment relevant to the child and family living with cerebral palsy using the International Classification of Functioning, Disability and Health for Children and Youth. J Rehabil Med. 2011;43(11):1010-5. This is a critical background study to the proposed study. In 239 children with cerebral palsy, “gait pattern functions” was identified 285 times as an outcome, outstripping by far all other outcomes. This gets to the irrelevance of the majority of outcome measures in this protocol. Parents are interested in gait and function, gait is not measured in this study and it should be. Short term changes in gait function can be reliably assessed without a control group, using a pre/post injection protocol. Other gold standard outcome measures such as GMFM are included and should be retained but changes will be difficult to interpret without a control group. Younger children are gaining in GMFM, and the gains in 12 months will outstrip any benefit from injection of BoNT-A. This is a necessary measure but can only be interpreted in an RCT format. Parents and clinicians will erroneously describe improvements in GMFM in a 4 year old child as the results of injection of BoNT-A combined with rehabilitation strategies, when it is in fact largely determined by natural history, as described by Rosenbaum et al Gross Motor Curves. Rosenbaum PL, Walter SD, Hanna SE, Palisano RJ, Russell DJ, Raina P, et al. Prognosis for gross motor function in cerebral palsy: creation of motor development curves. JAMA. 2002;288(11):1357-63. Outcome measures such as MTS have been discredited as being invalid, and unreliable. Interestingly, the Selective Motor Control Scale, which I devised with a co-author many years ago is referenced to a secondary publication.
--	--

	The CPQoL is a welcome addition to the study but will be difficult or impossible to interpret without better objective measures of gait and function. The Canadian Occupational Performance Measure (mCOPM) is in my view highly contentious. I have been present in clinic, when the Rehabilitation Physicians are establishing goals, later to be described as parent/child goals, for the injection episode. A frequent goal expressed by the parents of children at GMFCS Level II is “to be able to keep up with their peers”. Knowing that this is not possible, not deliverable, by any intervention currently available, the Researchers’ will offer a substitute goal such as “getting the heels closer to the floor”. Parents of children at GMFCS Level III will frequently state as a goal, that their child should be able to walk without the use of aids or assistive devices. Once again, knowing that this is not possible or deliverable, the Researchers’ will offer a substitute goal which is not the stated goal of the parents. I question the objectivity and scientific validity of mCOPM in a study such as this. Page 12, Lines 22 et seq: The primary and secondary outcome measures are well described and I have already alluded to the weaknesses of several of them. There are two major absences:  1. Given that the number one concern of parents for ambulant children is gait pattern, three-dimensional gait analysis is the sine qua non for this study. 2. Given the Authors’ reference to the harms of injection, listed in their Introduction, monitoring of patients for injection related harm is again in my view essential. The Authors’ propose using ultrasound to guide their injections. They should also be expected to use ultrasound to prospectively measure injection induced muscle atrophy and increase in connective tissue. Until recently, injections of Botulinum Toxin in children with cerebral palsy were considered to have small benefits but minimal or no harms. This promoted a culture encouraging injections to continue with ever increasing doses and decreasing intervals between injection. We are now in the situation where it is firmly established that injection of skeletal muscle causes short term harm and probably long-term harm. We are also aware that the benefits of injection are limited to modest short-term reductions in muscle tone with limited or no evidence for improvements in gait or function. As such it would be in my view, unethical not to measure the harms of injections, to compare them to the magnitude of benefit. Page 15, Lines 38 et seq: I have alluded to the subjective nature and the ability for investigators to manipulate the mCOPM. As such, it is not the best measure on which to base sample size calculation. Page 17, Lines 18-30: It would be most unwise to embark on this study, which includes children and adolescents in the age when orthopaedic surgery is most beneficial, without having an Orthopaedic surgeon as a member of the Steering Group, in order to “rescue” children from an ineffective therapy to something more beneficial. Lines 34-58: I do not believe that this is a novel study and I do not believe that the study design is capable of delivering scientifically valid or meaningful information.
--	---

VERSION 1 – AUTHOR RESPONSE

We would like to thank the reviewer for his comprehensive review of the manuscript and have responded to the individual points raised by the reviewer below.

Review of the Protocol for the Toxins Study: Understanding clinical and patient reported response of children and young people with cerebral palsy to intramuscular, lower limb Botulinum Toxin – A injections exploring all domains of the ICF: A pragmatic longitudinal observational study using a prospective one group repeated measures design.

Abstract:

Page 3, Lines 34-44: The influence of the ICF and ICF-CY have dominated recent research and many publications in the use of Botulinum Toxin in children with cerebral palsy. This is not a novel approach, and its value remains uncertain. Advantages include a more holistic view of the patient. Disadvantages include downgrading objective measures of gait and function in favour of subjective measures. A balance between objective and subjective outcomes is important.

The reviewer states that the use of ICF is not a novel approach and is of uncertain value. We would argue that whilst studies report body structure and function (BSF) and functional activity outcomes they rarely include participation and quality of life (QoL) outcomes, which have been identified as

most meaningful to children and families (1, 2)

Furthermore there is little evidence about the interaction between all domains of the ICF and we believe it is a novel approach to evaluate changes in all ICF domains simultaneously. The purpose of the current study is to examine the interaction (if any) between changes at BSF with other domains following injections. The authors agree that a balance between objective and subjective measures is important and as such, proposes to evaluate the effects of Botulinum Toxin across all domains

Methods and Analysis:

Page 3, Lines 45-55: This Reviewer considers that the Study Design (Pragmatic prospective longitudinal observational study) is a weak design and will be unable to provide scientific answers to the critical questions.

Mixed-methods research designs can be very powerful and are considered by many to be essential in the study of therapeutic interventions in CP (3). At no point have the authors stated that this is anything other than a pragmatic observational study evaluating real time established clinical practice which will have an effect on patient care over the next 5 years. It has been submitted as a protocol for such a study. It is not always possible to engage in RCTs in all fields and there is a place for pragmatic observational studies where this is the case (4). The protocol has been approved and funded in its current form by NIHR.

The changes in Gross Motor Function and gait function, as a result of gains in GMFM, in younger children, are an order of magnitude greater than the effects of injection of BoNT-A. As such parents and clinicians, will inevitably attribute gains in Gross Motor Function to the program of injections, when these are the result of the natural history.

The effect of natural history in CP is well acknowledged, but this does not mean that the investigation of the effects of interventions is redundant. The question of magnitude of change can be explained by changes in minimum clinically important differences (MCIDs) and minimal detectable changes (MDCs) with large and medium effect size using GMFM already reported in the literature (5, 6). In the present study, in order to specifically address this problem, all results will use published MCIDs in an attempt to evaluate the true effect of BoNT-A intervention.

Furthermore we know that by the age of 5 a typically developing child should score full marks on the GMFM. We recognise that there is a ceiling effect of GMFM as with many of the standardised clinical tests used to assess outcome following interventions in CP. The GMFM measures how much activity a child can do and not how well they can do it (i.e. the quality of the movement (QoM) activity). We have therefore included a further novel standardised assessment The Quality Function Measure

(QFM) to evaluate any change in QoM (positive or negative) following intervention.

Even in older children, when there are no gains in Gross Motor Function, the supportive environment of being in a clinic with supportive clinicians, with repeated assessments of the child, provides a powerful therapeutic environment, which is likely to yield positive responses in areas such as HRQoL. These effects are a response to the therapeutic environment plus the well-recognised placebo effect from injecting Botulinum Toxin. Without a control group, I think that it will not be possible to assess the magnitude of these confounding effects.

Botulinum toxin is now an accepted treatment in tone management in CP and therefore, due to ethical issues, many studies lack a control group (7). This is a pragmatic observational study aimed to evaluate current clinical practice within an established tertiary centre. We have acknowledged the limitation of a lack of a separate control group within the study. However, use of a pre and post design will allow each child to act as their own control. Furthermore the primary aim of this study is to interrogate whether any changes in BSF following injections can be related to a subsequent change in activity and participation, using a pre/post injection protocol.

Page 4, Lines 12-16: I do not believe that this is a novel study, there are numerous published studies which have employed the ICF Framework following Botulinum Toxin injection. In addition, some of these studies have a much stronger (RCT) design.

We assert that this is a novel study, the systematic review in progress by the authors (8) has identified a dearth of evidence investigating change in all aspects of the ICF framework following BoNT-A treatment and none to date have used the QFM. The authors reiterate that identifying the interaction following BoNT-A treatment within all domains of the ICF together with objectively measuring change in movement quality is the novel part of this study.

Lines 37-44: This is a serious weakness of the study. The SPACE BOP Study which separated out the

effect of Botulinum Toxin from the general rehabilitation milieu, in an RCT design, found no added benefit from injections.

We have acknowledged the variability of therapy provision as a confounder in our study. However we omitted the details of standardised care in our clinic, namely that in order to be accepted for injections all children are mandated to receive a weekly therapy block for 6 weeks post injections from their local services. Thank you for raising this we have now incorporated this in the manuscript. As this is a pragmatic study looking at current clinical practice, we will collect data regarding children's therapy programmes and plan to evaluate outcomes taking this into account. Schasfoort F, Dallmeijer A, Pangalila R, et al. Botulinum toxin has no added therapeutical value or cost-effectiveness for gross motor function, everyday physical activity or quality of life when combined with intensive functional physiotherapy. *Dev Med Child Neurol*, 2015;57(S5):8-9. This excellent study, has a stronger design than the proposed study in that it is a randomised controlled trial. It explores many of the outcome measures proposed in this study but in the RCT format.

The reviewer focuses on the SPACE BOP study, which following publication was shown to have a number of significant failings in study design. The quoted study attempted to use an RCT methodology but the authors themselves recognized that the randomisation process was flawed as children were allowed to change study arms due to parental pressure and acknowledged 'this was not a fully randomised trial' (9). Subsequently, the researchers have published an update paper concluding that 'several factors hindered the acceptance of their results' and retrospectively classified it as a 'pragmatically designed trial' rather than an RCT (10).

The authors when reflecting on the complexities of their SPACE BOP study suggest that in 'chronic (paediatric) rehabilitation populations, such as CP, for the future it is probably advisable to not only focus on randomised controlled trials'. This point is strongly reinforced in a recent editorial by Professor Peter Rosenbaum (3) when he suggests that mixed methods design rather than RCTs can be very powerful in this population and 'perhaps even essential'.

This study found no added benefit from injections of BoNT-A. However, the weaker study design proposed by these authors may well find a false positive result. This will confuse rather than clarify

the evidence base.

We argue that there is a gap in the research to carry out a pragmatic observational study where children receive 'usual care' with assessment tools that are widely available in the UK rather than intensive therapy and specialist analysis which is not always available outside of a research setting. As our study is not yet complete we cannot comment on whether the administration of BoNT-A results in positive or negative results within all of the ICF domains that we measure.

Introduction:

Page 5, Lines: 4-8: The crucial assertion that: "increased tone (hypertonia) is considered to be one of the primary motor impairments ..." is a strong and important statement. Significantly, there are no associated references. Clinicians focus too much on spasticity, when weakness, loss of selective motor

control and sensory disturbance, are of far greater importance to daily functioning, in children with cerebral palsy. than spasticity.

We acknowledge this oversight, apologies this should have been referenced (11, 12).

Lines 15-16: The idea that injection of BoNT-A is "reversible" is not correct, it is clearly not completely reversible. There are no animal or clinical studies in children to date which have convincingly demonstrated that the effects of a single injection of BoNT-A, ever wear off completely or that the injected muscle ever completely recovers.

We use the word 'reversible' in quotation marks and in line 19, page 4 discuss emerging evidence of long term pathological changes.

Lines 33-48: Rather than referencing, articles of variable scientific merit, the recent Cochrane Collaboration Report by Blumetti et al, would have been an excellent summary of the limitations of injection of BoNT-A. This report emphasises the small, temporary and inconsistent therapeutic effect, with poor to very poor supporting evidence.

We agree and are grateful for the reviewer for bringing this inadvertent omission to our attention.

Line 48: The "Traffic lights paper" has been criticised in the literature, and is not an adequate substitute for much more rigorous, unbiased and scientifically credible evidence summaries.

We do not disagree with the reviewers comments on the strength of this systematic review.

However the Novak paper is still widely quoted within the neurodisability community and remains the most frequently reproduced paper from DMCN over 6 years post-publication.

Lines 50-60: The Authors' list the weakness in many studies of BoNT-A treatment including single cycle of injections and short-term follow-up.

As referred to by the protocol authors, we agree that in the majority of studies referenced children were followed up short term usually between 12-16 weeks This is a strength of our pragmatic study following a child's usual care within the time of the study, participants will be followed up for twelve months.

Page 6, Lines 3-8: The Authors' are correct to indicate there are problems with current standardised outcome measures, stating that injections of BoNT-A fail to produce a positive change, at the level of one MCID. It is common practice to blame the outcome measure, when it is the intervention that is the weak point. The Authors' are not correct in stating that negative studies "focus predominantly on impairment measures". There are many studies using gold standard functional outcome measures including GMFM and 3-dimensional gait analysis, in which the response to BoNT-A injection is minimal or absent and below the MCID.

We would argue that children at GMFCS level I-II often reach the ceiling effect of GMFM, making it difficult to use this measure to evaluate change. We have therefore incorporated the QFM to further evaluate any change in movement quality.

Lines 11-23: This is a very important summary of the harmful effects of BoNT-A injection which need to be translated and incorporated in the Authors' protocol. We repeat at this point that there is no study, in animal models, or children with cerebral palsy which show that skeletal muscle ever recovers fully after a single injection of BoNT-A.

The authors strongly agree, and it is for this reason that side effect profile is incorporated into the protocol p 6 line 51. We are fully aware of the concerns regarding long term effects of BoNT-A and

the risk of overprescribing this treatment. However whilst important, it is beyond the scope of this study to resolve the issue as to whether skeletal muscle definitively does or does not recover from a single dose of BoNT-A.

Lines 25-37: The shifting emphasis from gold standard measures of gait and function, to “soft” outcome measures in activity measures and participation domains of ICF is regrettable. It reflects the negative outcomes of studies using the gold standard measures of function as much as a response to the prevailing orthodoxy that activities and participation are more important.

The authors reiterate that outcome measures are necessary in all domains of the ICF as detailed above. The reviewer repeatedly cites GMFM and gait analysis as the gold standard. However it is accepted that these measures do not address all the issues and functional changes that may occur. Improvements in gait analysis measures do not necessarily translate to improved participation (3). As such, we would suggest that rather than shifting the emphasis solely back to impairment measures that the inclusion of adaptive outcome measures which children and families value may add to the results of studies about the effectiveness (or ineffectiveness) of toxin intervention.

Lines 40-47: It is perfectly reasonable to suggest outcome measures in all domains of ICF but this should not be at the expense of ignoring the gold standard objective outcome measures.

The reviewer is once again suggesting that gait analysis is the only useful outcome tool in this field. Access to 3 DGA is limited in the majority of clinical settings. Therefore, as in our study, standardised proxy gait measures are often used such as 1 Minute Fast Walk Test (1MFWT), Modified Timed Up and Go (TUG) and Gross Motor Function Measure (GMFM:D&E dimensions). All of which can take place during clinical evaluation and are recognised as gold standard objective outcome measures.

Lines 50-58: This reviewer has mounting evidence that BoNT-A is overused to the detriment of longterm function in children with cerebral palsy. A single cycle of injections and study duration of 12 months is unable to address these key questions.

This pragmatic study aims to contribute to the knowledge base surrounding toxin use over a twelve month period (sometimes comprising two injection cycles) performed within a single site established practice. We have plans to follow up these children over a longer period beyond the study period as per our usual clinical practice.

Page 7, Lines 7-24: As stated above, this study has a weak design and will not achieve its desired goals of providing robust scientific evidence for the benefits versus harms for injection of BoNT-A. We believe a robust observational pragmatic study evaluating current clinical practice could provide useful information regarding a core clinical data set using a series of sensitive outcome measures related to toxin use. The aims of this study are not to provide the robust evidence alluded to by the reviewer and the goal of an RCT is not always achievable in this population (10). By combining quantitative and qualitative perspectives, we can gain understanding of the impacts of BoNT-A treatment that are likely to be both more credible and more pertinent than just change scores on one dependent variable such as 3DGA(3).

Line 57: The age range for inclusion is 4-18 years. We have previously reported no response to injections of BoNT-A in ambulant children with cerebral palsy, in a hemiplegic pattern by the age of 6 years because of the presence of fixed contracture.

It has been highlighted in previous studies that ‘age is not the primary prognostic indicator’ for BoNT-A use and some suggest it may be more about the stage of dynamic vs fixed contracture (13). Our clinical practice would suggest that it is the presence of a dynamic contracture rather than the absolute age of child which is a useful predictor in the appropriateness of a trial of toxin injections. The reviewer cites his own study in relation to hemiplegic children, however the participants in the present study consist of children within GMFCS levels I-III presenting with both unilateral and bilateral involvement

Hastings-Ison T, Sangeux M, Thomason P, Rawicki B, Fahey M, Graham HK. Onabotulinum toxin-A (Botox) for spastic equinus in cerebral palsy: a prospective kinematic study. *J Child Orthop*. 2018;12(4):390-7. In that study, children with diplegia had an “improvement” in equinus at the expense of increased crouch gait.

We welcome the addition of this study to the evidence base; particularly the investigation of two

injection cycle frequencies. As suggested by the authors, the concerns of overuse of toxin within this population may have been exacerbated by the practice of more frequent use of toxin in some centres. In some cases children are placed on a 'rolling programme' of treatment rather than waiting for the effects to wear off or be clinically indicated. Hastings-Ison and colleagues were rigorous in their aim to standardise practice, however they acknowledged that concomitant injection of the medial hamstrings may have presented as a confounder particularly in their study of children with diplegia. As with much research regarding BoNT-A the authors acknowledge 'there were both ethical and practical concerns regarding the inclusion of a no treatment control group.'

Older children and adolescents, with cerebral palsy GMFCS Levels I-III, have a complex mixture of weakness, spasticity, lower limb contractures and bony torsion which are not responsive in any meaningful way to injections of Botulinum Toxin A. Many of these children require orthopaedic surgery, yet the presence of contracture, torsion, or joint instability are not listed here as exclusions to being involved in the proposed study.

We acknowledge this point. Patients were selected for injections by the clinical team based on the presence of dynamic hypertonia in the muscle which limits function or causes pain. By using the term 'dynamic contracture' fixed contracture is implicitly excluded but this has been clarified further in the manuscript.

The protocol requires exclusions based on physical examination measures indicating the presence of muscle tendon contracture, bony torsion, and joint instability using clinical, radiographic and motion analysis data.

The reviewer is citing his own exclusion criteria. This is a study examining current practice and the clinical decision making process. This includes clinical examination including joint ROM, motion analysis with video gait analysis. As mentioned 3-DGA is not readily available to all and therefore is not used in our clinical setting

Lines 26-40: Much more detail regarding the injection protocol needs to be incorporated here, not just referral to reference 61.

This reference relates to therapy as previously stated, not injection protocol. The focus of our study is evaluation of toxin response rather than injection procedures which are standardised in our service and noted in the protocol.

Lines 45-55: The injection protocol is poorly described precluding replication or even understanding the connection between the indication and injection procedure.

This is a protocol, as such analysis of dose has not been completed nor selection of lower limb muscles injected. Given that the study is conducted within a single centre, we use a standardised dosing protocol derived from consensus use and manufacturers guidelines.

Lines 30 onwards: The Table describing the assessments and outcomes, has multiple subjective terms, precluding scientific analysis and replication.

We acknowledge this point and have changed this in the manuscript to improve clarity.

Page 10, Lines 10: For the first time, it is acknowledged that some children will develop a contracture requiring surgical intervention. The identification of these children and exclusion from this protocol needs to be clearly articulated. We now see a constant procession of children presenting to our Gait Analysis Laboratory, with decompensated musculoskeletal pathology, far too late for effective treatment, because of persisting with ineffective injections of BoNT-A over many cycles. To our knowledge there is not a definitive study comparing outcome following delayed surgery due to repeated use of toxin versus those children who receive surgery at an earlier age. The work by Molenaers and the Leuven group would suggest that the introduction of toxin injections delays both the age and the frequency of surgical procedures, which they argue is in the interest of the children (14). However longitudinal studies are required to evaluate the long term effects into adulthood (15) The authors suggest that 'both treatment modalities should be regarded as complimentary rather than mutually exclusive treatments, with both calling for an integrated approach'

Line 26: "Patient assessment takes between 60-90 minutes". This is burdensome for families for an intervention which the majority of current studies indicate will be ineffective.

The use of BoNT-A is widely accepted in management of muscle tone. There is no conclusive

evidence either of clear benefits or ill effects(16). The reviewer clearly has a very strong opinion in relation to the possible negative effects of toxin, but has failed to acknowledge that this is a study that may indicate either a positive or negative effect. We believe it is better to evaluate current clinical practice and rationalise future treatment based on the outcome of this pragmatic study for our clinical practice, since treatment with toxin remains an accepted treatment modality and forms part of international guidelines for management (17-19)

The reviewer goes on to mention that a 60 – 90 minute assessment is too burdensome for families. However the reviewer frequently suggests that 3D Gait Analysis should be used which takes on average 3 hours to perform for an individual child. Our usual assessments in contrast are 60-90 minutes and are carried out as part of our routine service, in which we provide advice on all aspects of tone management. We have highlighted the extra measures for the study are QFM which is scored at a later date with no extra burden on the families

Page 11: The ICF Table of Outcome Measures: This is a strange melange of outcome measures not all of which relate to real world concerns of the parents of ambulant children with cerebral palsy GMFCS Levels I-III. The majority of parents, whose children are ambulant, have concerns about their gait pattern and their Gross Motor Function.

Preston N, Clarke M, Bhakta B. Development of a framework to define the functional goals and outcomes of botulinum toxin A spasticity treatment relevant to the child and family living with cerebral palsy using the International Classification of Functioning, Disability and Health for Children and Youth. *J Rehabil Med.* 2011;43(11):1010-5.

This is a critical background study to the proposed study. In 239 children with cerebral palsy, “gait pattern functions” was identified 285 times as an outcome, outstripping by far all other outcomes. This gets to the irrelevance of the majority of outcome measures in this protocol. Parents are interested in gait and function, gait is not measured in this study and it should be. Short term changes in gait function can be reliably assessed without a control group, using a pre/post injection protocol

Outcomes in all aspects of ICF will be measured. Undoubtedly change in gait function is important, however this does not exclude the addition of other goals of treatment as identified by children and families such as pain and integration into community life (20, 21). We are yet to see whether our goal setting data corroborates this or introduces new SMART activity and participation goals. This study uses proxy gait measures TUG, 1MFWT and GMFM (D& E) which are commonly used in clinical

practice and have been shown to be valid indicators of motor function in CP (5). Our intention is to embed research within clinical practice and not remove it from a clinical setting.

Other gold standard outcome measures such as GMFM are included and should be retained but changes will be difficult to interpret without a control group. Younger children are gaining in GMFM, and the gains in 12 months will outstrip any benefit from injection of BoNT-A. This is a necessary measure but can only be interpreted in an RCT format. Parents and clinicians will erroneously describe improvements in GMFM in a 4 year old child as the results of injection of BoNT-A combined with rehabilitation strategies, when it is in fact largely determined by natural history, as described by Rosenbaum et al Gross Motor Curves.

Rosenbaum PL, Walter SD, Hanna SE, Palisano RJ, Russell DJ, Raina P, et al. Prognosis for gross motor

function in cerebral palsy: creation of motor development curves. *JAMA.* 2002;288(11):1357-63.

The GMFM is well validated for use in this population and we propose to relate all results to published MCIDs. This allows any additional effects of treatment to be separated from natural change due to maturation.

Outcome measures such as MTS have been discredited as being invalid, and unreliable.

It is well recognised that there are issues with robust clinical spasticity measures in this field, with a lack of instrumented spasticity measures available for use in the clinical setting (22) Nevertheless in the absence of more validated measures the majority of BoNT-A studies continue to report the use of MTS or MAS to clinically measure tone. Whilst the authors acknowledge their limitations it allows

the use of a common language with reference to standard definitions (23) for other workers in the field to compare results.

The CPQoL is a welcome addition to the study but will be difficult or impossible to interpret without better objective measures of gait and function.

QFM is our primary outcome and as it is the first published use in this field we are yet to measure how it relates to outcome of proxy measures. As we are using gold standard measures of GMFM TUG and 1 MFWT we believe we have objective measures to assist in our interpretation of the data.

The Canadian Occupational Performance Measure (mCOPM) is in my view highly contentious. I have been present in clinic, when the Rehabilitation Physicians are establishing goals, later to be described as parent/child goals, for the injection episode. A frequent goal expressed by the parents of children at GMFCS Level II is "to be able to keep up with their peers". Knowing that this is not possible, not deliverable, by any intervention currently available, the Researchers' will offer a substitute goal such as "getting the heels closer to the floor".

Parents of children at GMFCS Level III will frequently state as a goal, that their child should be able to walk without the use of aids or assistive devices. Once again, knowing that this is not possible or deliverable, the Researchers' will offer a substitute goal which is not the stated goal of the parents. I question the objectivity and scientific validity of mCOPM in a study such as this.

COPM is widely used as an outcome measure in Neurodisability research and frequently used in clinical practice (24). Like all proxy outcome measures, use of the COPM depends on the expertise of those administering the tool. The skill is in allowing parents and children to identify their own goals around Occupational Performance Issues (OPIs) and set SMART goals which they hope to see change

following lower limb botulinum toxin injections. We hope that the results of this study will provide useful information regarding the use of QFM the novel primary outcome measure in future sample size calculations.

Page 12, Lines 22 et seq: The primary and secondary outcome measures are well described and I have already alluded to the weaknesses of several of them. There are two major absences:

1. Given that the number one concern of parents for ambulant children is gait pattern, threedimensional gait analysis is the sine qua non for this study.

As mentioned above, we have used proxy gait measures (1MFWT/TUG) which some would argue more closely reflect every day walking activities and functional activities.

2. Given the Authors' reference to the harms of injection, listed in their Introduction, monitoring of patients for injection related harm is again in my view essential. The Authors' propose using ultrasound to guide their injections. They should also be expected to use ultrasound to prospectively measure injection induced muscle atrophy and increase in connective tissue.

Whilst the authors acknowledge the role of 3D ultrasound to evaluate toxin muscle atrophy, this is beyond the scope of this clinical study. Adverse events are routinely recorded.

Until recently, injections of Botulinum Toxin in children with cerebral palsy were considered to have small benefits but minimal or no harms. This promoted a culture encouraging injections to continue with ever increasing doses and decreasing intervals between injection. We are now in the situation where it is firmly established that injection of skeletal muscle causes short term harm and probably long-term harm. We are also aware that the benefits of injection are limited to modest short-term reductions in muscle tone with limited or no evidence for improvements in gait or function. As such it would be in my view, unethical not to measure the harms of injections, to compare them to the magnitude of benefit.

We would like to emphasise that this study has been reviewed approved and funded by NIHR and has both internal and external ethical approval. We would never knowingly act in a way that we felt was harmful to the children in our care and the administration of BoNT-A is within the scope of national and international guidelines (16-18). Adverse events are routinely recorded. In our view, with respect, the reviewer is digressing into issues outside the scope of this study. It would indeed be possible to investigate many of the effects of this intervention, but as highlighted by the large and controversial body of evidence regarding the use of BoNT-A it is not possible to address all these

issues in a single study.

Whilst the reviewer himself has grave concerns about the use and long term effects of BoNT-A injections in CYP with CP and is keen to expound his opinion, this view is not shared by all working in Neurodisability (25). We believe further research is required to optimise the selection of patients who may benefit from this treatment and also provide information regarding those patients where BoNT-A may not be the correct treatment option. This mixed methods study, assessing outcome over a twelve month period aims to evaluate change in ICF outcomes following a change in BSF for our patient group, investigating any change in function and QoM in order to optimise the targeting of treatment.

Page 17, Lines 18-30: It would be most unwise to embark on this study, which includes children and adolescents in the age when orthopaedic surgery is most beneficial, without having an Orthopaedic surgeon as a member of the Steering Group, in order to “rescue” children from an ineffective therapy to something more beneficial.

We believe this to be rather an emotive statement and with respect, could be considered by many to be disrespectful to the multiple specialists involved in the care of the children in our service. Children up to the age of 18 years are seen in an integrated neurodisability service within a tertiary children’s hospital, as such we hold regular combined clinics with our Orthopaedic and Neurosurgical colleagues and cross refer children between our services as required. We maintain that not all children receiving BoNT-A require orthopaedic intervention.

In a recent publication highlighting the potential harm of toxin co-authored by the reviewer (26) the assumption is made that ‘almost 100%’ of children receiving toxin later proceed to orthopaedic surgery. However, this claim has been challenged, data from Western Australia State Paediatric Rehabilitation Service has shown that less than 20% of children with GMFCS level I-III receiving BoNT-A injections require surgery by the age of 10 (23).

Lines 34-58: I do not believe that this is a novel study and I do not believe that the study design is capable of delivering scientifically valid or meaningful information.

The reviewer is well known in the field but similarly is known to have recently adopted very strong views against the use of Botulinum Toxin A (BoNT-A) in Cerebral Palsy (CP). Whilst the reviewer makes a number of valid points which we welcome and have addressed in the manuscript, he has failed to acknowledge that fundamentally this is a protocol to explore the effects of current practice in a tertiary centre. It is not seeking to justify its use but to comprehensively describe its effects in ambulant children with CP. In criticising this protocol, he selectively cites studies regarding the use of BoNT-A which have negative findings without acknowledging other points of view. We do not accept that this precludes further studies in this field. The reviewer makes his views very forcefully and we have attempted to address his points in the text above. It is clear that he has grave reservations as to the use of any BoNT-A in CYP with CP.

As we have stressed, BoNT-A is still an accepted part of the management of hypertonia in children with CP within many Neurodisability and Orthopaedic centres. We believe that this study is novel in its use of evaluating all aspects of the ICF domain along with the introduction of QFM an objective measure to evaluate Quality of Movement. We expect that the results of this study will contribute to the knowledge base informing the use of BoNT-A in this patient group and assist in the development of a core data set of clinical outcomes to optimise service delivery including discontinuing its use when not appropriate and permit further study in this area.

References

1. Jette AM, Haley SM. Contemporary measurement techniques for rehabilitation outcomes assessment. *J Rehabil Med.* 2005;37(6):339-45.
2. Gordon AL. Functioning and disability after stroke in children: using the ICF-CY to classify health outcome and inform future clinical research priorities. *Dev Med Child Neurol.* 2014;56(5):434-44.
3. Rosenbaum P. How do we know if interventions in developmental disability are effective?

- Developmental Medicine & Child Neurology. 2020;62(12):1344-.
4. Bussmann JBJ, Pangalila RF, Stam HJ, Schasfoort FC. THE ROLE OF BOTULINUM TOXIN IN MULTIMODAL TREATMENT OF SPASTICITY IN AMBULATORY CHILDREN WITH SPASTIC CEREBRAL PALSY: EXTENSIVE EVALUATION OF A COST-EFFECTIVENESS TRIAL. *Journal of Rehabilitation Medicine*. 2020;52(5).
 5. Oeffinger D, Bagley A, Rogers S, Gorton G, Kryscio R, Abel M, et al. Outcome tools used for ambulatory children with cerebral palsy: responsiveness and minimum clinically important differences. *Dev Med Child Neurol*. 2008;50.
 6. Wang HY, Yang YH. Evaluating the responsiveness of 2 versions of the gross motor function measure for children with cerebral palsy. *Arch Phys Med Rehabil*. 2006;87(1):51-6.
 7. Hastings-Ison T, Blackburn C, Rawicki B, Fahey M, Simpson P, Baker R, et al. Injection frequency of botulinum toxin A for spastic equinus: a randomized clinical trial. *Developmental medicine and child neurology* [Internet]. 2016; 58(7):[750-7 pp.]. Available from: <http://onlinelibrary.wiley.com/doi/10.1111/dmnc.12962>
 8. Katchburian L CL, Coghill J, Chugh D, Crowe B, Oulton K. What are the effects of lower limb botulinum toxin A injections on activity, participation and quality of life in ambulant children with cerebral palsy? 2019.
 9. Schasfoort F, Dallmeijer A, Pangalila R, Catsman C, Stam H, Becher J, et al. Value of botulinum toxin injections preceding a comprehensive rehabilitation period for children with spastic cerebral palsy: A cost-effectiveness study. *Journal of Rehabilitation Medicine*. 2018;50(1):22-9.
 10. Bussmann JBJ, Pangalila RF, Stam HJ, Schasfoort F. The role of botulinum toxin in multimodal treatment of spasticity in ambulatory children with spastic Cerebral Palsy: extensive evaluation of a cost-effectiveness trial. *Journal of Rehabilitation Medicine*. 2020;52(5):jrm00059.
 11. Koman LA, Paterson Smith B, Balkrishnan R. Spasticity associated with cerebral palsy in children: guidelines for the use of botulinum A toxin. *Paediatric Drugs*. 2003;5(1):11-23.
 12. Koman LA, Mooney JF, 3rd, Smith BP, Goodman A, Mulvaney T. Management of spasticity in cerebral palsy with botulinum-A toxin: report of a preliminary, randomized, double-blind trial. *Journal of Pediatric Orthopedics*. 1994;14(3):299-303.
 13. Eames NWA, Baker R, Hill N, Graham K, Taylor T, Cosgrove A. The effect of botulinum toxin A on gastrocnemius length: Magnitude and duration of response. *Developmental Medicine and Child Neurology*. 1999;41(4):226-32.
 14. Molenaers G, Van Campenhout A, Fagard K, De Cat J, Desloovere K. The use of botulinum toxin A in children with cerebral palsy, with a focus on the lower limb. *J Child Orthop*. 2010;4.
 15. Molenaers G, Desloovere K, Fabry G, De Cock P. The effects of quantitative gait assessment and botulinum toxin A on musculoskeletal surgery in children with cerebral palsy. *Journal of Bone and Joint Surgery-American Volume*. 2006;88A(1):161-70.
 16. Blumetti FC, Belloti JC, Tamaoki MJS, Pinto JA. Botulinum toxin type A in the treatment of lower limb spasticity in children with cerebral palsy. *Cochrane Database of Systematic Reviews*. 2019(10).
 17. NICE. Spasticity in children and young people with non progressive brain disorders. *Clinical Guideline 145*; . London: National Institute for Health and Clinical Excellence. ; 2012.
 18. Love SC, Novak I, Kentish M, Desloovere K, Heinen F, Molenaers G, et al. Botulinum toxin assessment, intervention and after-care for lower limb spasticity in children with cerebral palsy: international consensus statement. *European Journal of Neurology*. 2010;17 Suppl 2:9-37.
 19. Heinen F, Desloovere K, Schroeder AS, Berweck S, Borggraefe I, van Campenhout A, et al. The updated European Consensus 2009 on the use of Botulinum toxin for children with cerebral palsy. *European Journal of Paediatric Neurology*. 2010;14(1):45-66.
 20. Vargus-Adams JN, Martin LK. Domains of importance for parents, medical professionals and youth with cerebral palsy considering treatment outcomes. *Child: care, health and development*. 2011;37(2):276-81.

21. Valentine J, Davidson S-A, Bear N, Blair E, Paterson L, Ward R, et al. A prospective study investigating gross motor function of children with cerebral palsy and GMFCS level II after long-term Botulinum toxin type A use. *BMC pediatrics*. 2020;20(1):7-.
22. Bar-On L, Van Campenhout A, Desloovere K, Aertbelien E, Huenaerts C, Vandendooren B, et al. Is an instrumented spasticity assessment an improvement over clinical spasticity scales in assessing and predicting the response to integrated botulinum toxin type a treatment in children with cerebral palsy? *Archives of Physical Medicine & Rehabilitation*. 2014;95(3):515-23.
23. Graham HK, Rosenbaum P, Paneth N, Dan B, Lin J-P, Damiano DL, et al. Cerebral palsy. *Nature Reviews Disease Primers*. 2016;2(1):15082.
24. Kang M, Smith E, Goldsmith CH, Switzer L, Rosenbaum P, Wright FV, et al. Documenting change with the Canadian Occupational Performance Measure for children with cerebral palsy. *Developmental Medicine and Child Neurology*. 2020;62(10):1154-60.
25. Langdon K, Copeland L, Edwards P, Rodwell K, McLennan K, Carroll T, et al. Comment on: "Botulinum Toxin in the Management of Children with Cerebral Palsy". *Pediatric Drugs*. 2019.
26. Multani I, Manji J, Hastings-Ison T, Khot A, Graham K. Botulinum Toxin in the Management of Children with Cerebral Palsy. *Paediatric drugs*. 2019;21(4):261-81.